# Identification and Tracking of Antiviral Drug Combinations

**DOI:** 10.3390/v12101178

**Published:** 2020-10-18

**Authors:** Aleksandr Ianevski, Rouan Yao, Svetlana Biza, Eva Zusinaite, Andres Männik, Gaily Kivi, Anu Planken, Kristiina Kurg, Eva-Maria Tombak, Mart Ustav, Nastassia Shtaida, Evgeny Kulesskiy, Eunji Jo, Jaewon Yang, Hilde Lysvand, Kirsti Løseth, Valentyn Oksenych, Per Arne Aas, Tanel Tenson, Astra Vitkauskienė, Marc P. Windisch, Mona Høysæter Fenstad, Svein Arne Nordbø, Mart Ustav, Magnar Bjørås, Denis E. Kainov

**Affiliations:** 1Department of Clinical and Molecular Medicine, Norwegian University of Science and Technology, 7028 Trondheim, Norway; aleksandr.ianevski@ntnu.no (A.I.); rouany@stud.ntnu.no (R.Y.); svetlana.biza@yandex.ru (S.B.); hilde.lysvand@ntnu.no (H.L.); kirsti.loseth@ntnu.no (K.L.); valentyn.oksenych@ntnu.no (V.O.); per.a.aas@ntnu.no (P.A.A.); Mona.Hoyseter.Fenstad@stolav.no (M.H.F.); svein.a.nordbo@ntnu.no (S.A.N.); magnar.bjoras@ntnu.no (M.B.); 2Institute of Technology, University of Tartu, 50090 Tartu, Estonia; eva.zusinaite@ut.ee (E.Z.); nastassia.shtaida@ut.ee (N.S.); tanel.tenson@ut.ee (T.T.); 3Icosagen Cell Factory OÜ, 61713 Kambja vald Tartumaa, Estonia; andres.mannik@icosagen.ee (A.M.); gaily.kivi@icosagen.ee (G.K.); anu.planken@icosagen.ee (A.P.); kristiina.kurg@icosagen.ee (K.K.); eva-maria.tombak@icosagen.ee (E.-M.T.); mart.ustav.jr@icosagen.com (M.U.J.); mart.ustav@icosagen.ee (M.U.); 4Institute for Molecular Medicine Finland, FIMM, University of Helsinki, 00014 Helsinki, Finland; evgeny.kulesskiy@helsinki.fi; 5Applied Molecular Virology Laboratory, Institut Pasteur Korea, Sampyeong-dong 696, Bundang-gu, Seongnam-si 463-400, Gyeonggi-do, Korea; eunji.jo@ip-korea.org (E.J.); jaewon.yang@ip-korea.org (J.Y.); marc.windisch@ip-korea.org (M.P.W.); 6Department of Laboratory Medicine, Lithuanian University of Health Science, 44307 Kaunas, Lithuania; astra.vitkauskiene@kaunoklinikos.lt; 7Department of Medical Microbiology, St. Olavs Hospital, 7006 Trondheim, Norway; 8Department of Immunology and Transfusion Medicine, St. Olavs Hospital, 7006 Trondheim, Norway

**Keywords:** antivirals, antiviral drug combinations, broad-spectrum antivirals, virus

## Abstract

Combination therapies have become a standard for the treatment for HIV and hepatitis C virus (HCV) infections. They are advantageous over monotherapies due to better efficacy, reduced toxicity, as well as the ability to prevent the development of resistant viral strains and to treat viral co-infections. Here, we identify new synergistic combinations against severe acute respiratory syndrome coronavirus 2 (SARS-CoV-2), echovirus 1 (EV1), hepatitis C virus (HCV) and human immunodeficiency virus 1 (HIV-1) in vitro. We observed synergistic activity of nelfinavir with convalescent serum and with purified neutralizing antibody 23G7 against SARS-CoV-2 in human lung epithelial Calu-3 cells. We also demonstrated synergistic activity of nelfinavir with EIDD-2801 or remdesivir in Calu-3 cells. In addition, we showed synergistic activity of vemurafenib with emetine, homoharringtonine, anisomycin, or cycloheximide against EV1 infection in human lung epithelial A549 cells. We also found that combinations of sofosbuvir with brequinar or niclosamide are synergistic against HCV infection in hepatocyte-derived Huh-7.5 cells, and that combinations of monensin with lamivudine or tenofovir are synergistic against HIV-1 infection in human cervical TZM-bl cells. These results indicate that synergy is achieved when a virus-directed antiviral is combined with another virus- or host-directed agent. Finally, we present an online resource that summarizes novel and known antiviral drug combinations and their developmental status.

## 1. Introduction

Every year, emerging and re-emerging viruses, such as severe acute respiratory syndrome coronavirus 2 (SARS-CoV-2), Middle East respiratory syndrome (MERS-CoV), Zika virus (ZIKV), Ebola virus (EBOV), influenza A virus (FLUAV), and Rift Valley fever virus (RVFV) surface from natural reservoirs to infect, disable, and kill people [1,2]. As of October 2020, the number of people infected with SARS-CoV-2 continues to rise, with a global death toll of more than 1 million.

Drug combinations are emerging as useful tools to treat infections because of their increased efficacy and decreased toxicity. Additionally, combinations of two or more antivirals could be administered in a cocktail to target a broad range of viruses or to prevent the development of drug-resistance, a common consequence of antiviral monotherapy. For these reasons, antiviral drug combinations could serve as front line therapeutic options against poorly characterized emerging viruses, re-emerging drug-resistant viral strains or viral co-infections.

Indeed, antiviral drug combinations have become a standard treatment of rapidly evolving viruses, such as HIV and hepatitis C virus (HCV) [3,4]. These include abacavir/dolutegravir/lamivudine (triumeq), darunavir/cobicistat/emtricitabine/tenofovir (symtuza), ledipasvir/sofosbuvir, sofosbuvir/velpatasvir, and lopinavir/ritonavir (kaletra). Many other drug combinations are currently in clinical trials against SARS-CoV-2, HCV, HBV, HSV-1, and other viral infections (ChiCTR2000029308, NCT04291729, ChiCTR2000030894, NCT03111108, NCT01045278, NCT00255034, NCT02480166, NCT00383864, NCT01023217, NCT00922207, NCT02360592, NCT00922207, etc.).

The individual components of novel antiviral combinations often come from the pool of approved and investigational drugs that have already passed phase I clinical trials (Figure 1). From there, in vitro studies can uncover synergism among drug combinations. Once proof-of-concept has been established, repositioned drugs can enter preclinical and clinical trials. Depending on the existing body of knowledge around the drug components, these combinations may also directly enter phase I–II clinical studies in cases of outstanding urgency, allowing for a cheaper and faster path to market.

Our recent in vitro studies have revealed antiviral synergism across a wide range of compounds and viruses. We have found that the combination of pimodivir and gemcitabine shows synergism against FLUAV infection in human macrophages [5]. We have also demonstrated that the combination of obatoclax and saliphenylhalamide is synergistic against ZIKV infection in retinal pigment epithelium (RPE) cells [6]. We have also shown that combinations of nelfinavir with salinomycin, amodiaquine, obatoclax, emetine, and homoharringtonine all exhibit synergistic activity against SARS-CoV-2 in kidney epithelial cells extracted from an African green monkey (Vero-E6) [7]. These and other studies suggest that synergy can be achieved when virus-directed agent is combined with another virus-directed or host-directed agent, while combinations of host-directed agents often have toxic consequences in cell culture.

The goal for this study was to uncover novel synergistic drug combinations by combining virus-directed agent with another virus- or host-directed antivirals. We found that combinations of nelfinavir with convalescent serum, neutralizing antibody, EIDD-2801 or remdesivir were synergistic against SARS-CoV-2 infection in human non-small cell lung cancer Calu-3 cells. Further, combinations of vemurafenib together with emetine, homoharringtonine, anisomycin and cycloheximide were found to be synergistic against EV1 infection in A549 cells. Combinations of sofosbuvir with brequinar and niclosamide were shown to be effective against HCV infection in Huh-7.5 cells. Finally, combinations of monensin with lamivudine and tenofovir were shown to have antiviral synergy against HIV-1 infection in TZM-bl cells. We also present a freely accessible web resource that aggregates known synergistic interactions between antiviral agents.

## 2. Materials and Methods

### 2.1. Drugs

Appendix A lists compounds, their suppliers, and catalogue numbers. To obtain 10 mM stock solutions, compounds were dissolved in dimethyl sulfoxide (DMSO; Sigma-Aldrich, Hamburg, Germany) or milli-Q water. Collection of convalescent serum from patients recovered from COVID-19 (SARS-CoV-2_2) has been described in our previous study [7]. The identification, cloning, expression and purification of human neutralizing antibodies are described elsewhere (“SARS CoV-2 Neutralizing Antibodies”, patent application number: 63090576). The solutions were stored at −80 °C until use.

### 2.2. Cell Cultures

Calu-3 and RPE cells were grown in DMEM-F12 supplemented with 10% FBS, 100 µg/mL streptomycin, and 100 U/mL penicillin (Pen–Strep). Human adenocarcinoma alveolar basal epithelial A549 and Vero-E6 cells were grown in DMEM supplemented with 10% FBS and Pen–Strep. The cell lines were maintained at 37 °C with 5% CO_2_. ACH-2 cells, which possess a single integrated copy of the provirus HIV-1 strain LAI (NIH AIDS Reagent Program), were grown in RPMI-1640 medium supplemented with 10% FBS and Pen–Strep. TZM-bl, previously designated JC53-bl (clone 13) is a human cervical cancer HeLa cell line, stably expressing the firefly luciferase under control of the HIV-1 LTR promoter. TZM-bl cells were grown in DMEM supplemented with 10% FBS and Pen/Strep. The human hepatoma Huh-7.5 cell line was grown in DMEM supplemented with 10% FBS, non-essential amino acids (NEAA), L-glutamine and Pen–Strep [8]. All cell lines were grown in a humidified incubator at 37 °C in the presence of 5% CO_2_.

### 2.3. Viruses

The SARS-CoV-2 hCoV-19/Norway/Trondheim-S15/2020 strain has been described in our previous study [7]. It was amplified in a monolayer of Vero-E6 cells in the DMEM media containing Pen–Strep and 0.2% bovine serum albumin. EV1 (Farouk strain; ATCC) was provided by Prof. Marjomäki from University of Jyväskylä. EV6 was isolated in our laboratory [9]. EV1 and EV6 viruses were amplified in a monolayer of A549 cells in the DMEM media containing Pen–Strep and 0.2% bovine serum albumin (BSA). To produce HIV-1, 6 × 10^6^ ACH-2 cells were seeded in 10 mL medium. Virus production was induced by the addition of 100 nM phorbol-12-myristate-13-acetate. The cells were incubated for 48 h, and the HIV-1-containing medium was collected. The amount of HIV-1 was estimated by measuring the concentration of HIV-1 p24 in the medium using anti-p24-ELISA, which was developed in-house. Recombinant purified p24 protein was used as a reference. Cell culture-derived infectious HCV (HCVcc) was produced as described before [10]. Briefly, Huh-7.5 cells transiently transfected with HCV RNA transcripts of a cell culture-adapted JFH1 genome expressing NS5A-GFP fusion protein (JFH1_5 A/5B_GFP). HCVcc containing medium was collected at 4 days post-transfection. Viral supernatant was clarified by filtration using a syringe filter with a 0.2 µm pore size (Millipore, Bedford, MA, USA). All virus stocks were stored at −80 °C.

### 2.4. Neutralization Assay

Approximately 4 × 10^4^ Vero-E6 cells were seeded per well in 96-well plates. The cells were grown for 24 h in DMEM supplemented with 10% FBS and Pen-Strep. Serum sample was prepared in 3-fold dilutions at 7 different concentrations, starting from 40 µg/mL in the virus growth medium (VGM) containing 0.2% BSA and Pen–Strep in DMEM. Virus hCoV-19/Norway/Trondheim-S15/2020 was added to achieve a multiplicity of infection (moi) of 0.1 and incubated for 1h at 37 °C. 0.1% DMSO was added to the control wells. The Vero-E6 cells were overplayed with VGM containing mixture of the virus and convalescent serum. After 72 h the medium was removed, and a CellTiter-Glo assay (Promega, Oslo, Norway) was performed to measure cell viability.

### 2.5. Drug Test

Approximately 4 × 10^4^ Vero-E6, A549, or RPE cells were seeded per well in 96-well plates. The cells were grown for 24 h in DMEM or DMEM-F12 supplemented with 10% FBS and Pen–Strep. The medium was replaced with DMEM or DMEM-F12 containing 0.2% BSA and Pen–Strep. The compounds were added to the cells in 3-fold dilutions at 7 different concentrations, starting from 30 µM. No compounds were added to the control wells. The cells were mock- or virus-infected at a moi of 0.1. After 24 (EV1) or 72 (SARS-CoV-2) h of infection, the medium was removed from the cells, and a CellTiter-Glo assay was performed to measure cell viability.

The half-maximal cytotoxic concentration (CC_50_) for each compound was calculated based on viability/death curves obtained on mock-infected cells after nonlinear regression analysis with a variable slope using GraphPad Prism software version 7.0a (GraphPad Software, San Diego, CA, USA). The half-maximal effective concentrations (EC_50_) were calculated based on the analysis of the viability of infected cells by fitting drug dose–response curves using the four-parameter (4PL) logistic function *f*(*x*):
(1)$$f\left(x\right)=A_{min}+\frac{A_{max}-A_{min}}{1+{\left(\frac{x}{m}\right)}^{\lambda }},$$
where *f*(*x*) is a response value at dose *x*, *A_min_* and *A_max_* are the upper and lower asymptotes (minimal and maximal drug effects), *m* is the dose that produces the half-maximal effect (EC_50_ or CC_50_), and *λ* is the steepness (slope) of the curve. The relative effectiveness of the drug was defined as the selectivity index (SI = CC_50_/EC_50_).

### 2.6. Virus Quantification

For testing the production of infectious virions, we titered the viruses as described in our previous studies [11,12,13]. In summary, media from the viral culture were serially diluted from 10^−2^ to 10^−7^ in serum-free media containing 0.2% BSA. The dilutions were applied to a monolayer of Vero-E6 (for SARS-CoV-2) or A549 (for EV1) cells in 24-well plates. After one hour, cells were overlaid with virus growth medium containing 1% carboxymethyl cellulose and incubated for 72 (for SARS-CoV-2) or 48 h (for EV1). The cells were fixed and stained with crystal violet dye, and the plaques were calculated in each well and expressed as plaque-forming units per mL (pfu/mL).

### 2.7. Drug Combination Test and Synergy Calculations

Vero-E6 cells were treated with different concentrations of two drugs and infected with SARS-CoV-2 (moi 0.1) or mock. After 72 h, cell viability was measured using CellTiter-Glo. A549 cells were treated with different concentrations of two drugs and infected with EV1 (moi 0.1) or mock. After 24 h, cell viability was measured using CellTiter-Glo. TZM-bl cells were treated with different concentrations of two drugs and infected with HIV-1 (corresponding to 300 ng/mL of HIV-1 p24) or mock. After 48 h post infection (hpi), the media was removed from the cells, the cells were lysed, and firefly luciferase activity was measured using the luciferase assay system (Promega, Madison, WI, USA). In a parallel experiment, cell viability was measured using CellTiter-Glo. We also examined toxicity and antiviral activity of drug combinations using GFP-expressing HCV in Huh-7.5 cells by following previously described procedures [10].

To test whether the drug combinations act synergistically, the observed responses were compared with expected combination responses. The expected responses were calculated based on the ZIP reference model using SynergyFinder version 2 [14,15]. We quantified synergy scores, which represent the average excess response due to drug interactions (i.e., 10% of cell survival beyond the expected additivity between single drugs has a synergy score of 5).

### 2.8. Gene Expression Analysis

A549 cells were treated with 10 µM vemurafenib or vehicle at indicated concentrations. Cells were infected with EV1 at moi 0.1 or mock. After 8 h, total RNA was isolated using RNeasy Plus Mini kit (Qiagen, Hilden, Germany). Gene expression profiling was done using Human HT-12 v4 Expression BeadChip Kit (Illumina, San Diego, CA, USA) according to the manufacturer’s recommendation as described previously [5]. Raw microarray data were normalized using the BeadArray and Limma packages from Bioconductor suite for R [16]. Normalized data were further processed using variance and intensity filter. Genes differentially expressed between samples and controls were determined using the Limma package. Benjamini–Hocberg multiple correction testing method was used to filter out differentially expressed genes based on a q-value threshold (*q* < 0.05). Filtered data were sorted by logarithmic fold change (log2FC). Heatmap was generated using an in-house developed interface, Breeze [17]. Gene set enrichment analysis was performed using open-source software (www.broadinstitute.org/gsea).

### 2.9. Cytokine Profiling

The medium from EV1- or mock-infected, non- or drug-treated A549-cells was collected at 24 hpi and clarified by centrifugation for 5 min at 14,000 rpm. Cytokines were analyzed using Proteome Profiler Human Cytokine Array Kit (R&D Systems, Abingdon, UK) according to the manufacturer’s instructions.

### 2.10. Metabolic Analysis

Metabolomics analysis was performed as described previously [5]. Briefly, 10 µL of labeled internal standard mixture was added to 100 µL of the sample (cell culture media). Next, 0.4 mL of solvent (99% ACN and 1% FA) was added to each sample. The insoluble fraction was removed by centrifugation (14,000 rpm, 15 min, 4 °C). The extracts were dispensed in OstroTM 96-well plate (Waters Corporation, Milford, MA, USA) and filtered by applying vacuum at a delta pressure of 300–400 mbar for 2.5 min on Hamilton StarLine robot’s vacuum station. The clean extract was collected to a 96-well collection plate, placed under the OstroTM plate. The collection plate was sealed and centrifuged for 15 min, 4000 rpm, 4 °C, and placed in autosampler of the liquid chromatography system for the injection.

Sample analysis was performed on an Acquity UPLC-MS/MS system (Waters Corporation, Milford, MA, USA). The autosampler was used to perform partial loop with needle overfill injections for the samples and standards. The detection system, a Xevo TQ-S tandem triple quadrupole mass spectrometer (Waters, Milford, MA, USA), was operated in both positive and negative polarities with a polarity switching time of 20 ms. Electrospray ionization (ESI) was chosen as the ionization mode with a capillary voltage at 0.6 KV in both polarities. The source temperature and desolvation temperature of 120 °C and 650 °C, respectively, were maintained constant throughout the experiment. Declustering potential (DP) and collision energy (CE) were optimized for each compound. Multiple reaction monitoring (MRM) acquisition mode was selected for quantification of metabolites with individual span time of 0.1 s given in their individual MRM channels. The dwell time was calculated automatically by the software based on the region of the retention time window, number of MRM functions and also depending on the number of data points required to form the peak. MassLynx 4.1 software was used for data acquisition, data handling, and instrument control (Agilent, Santa Clara, CA, USA).

Data processing was done using TargetLynx software (Agilent, Santa Clara, CA, USA), and metabolites were quantified by calculating curve area ratio using labeled internal standards (IS) (area of metabolites/area of IS) and external calibration curves. Metabolomics data were log_2_ transformed for linear modeling and empirical-Bayes-moderated *t*-tests using the LIMMA package (https://bioconductor.org/packages/release/bioc/html/limma.html). To analyze the differences in metabolite levels, a linear model was fit for each metabolite. The Benjamini–Hochberg method was used to correct for multiple testing. The significant metabolites were determined at a Benjamini–Hochberg false discovery rate (FDR) controlled at 10%. The heatmap was generated using the pheatmap package (https://cran.rproject.org/web/packages/pheatmap/index.html) based on log-transformed profiling data. MataboAnalyst 3.0 was used to identify pathways related to EV1 infection (www.msea.ca). In this pathway analysis tool, the pathway data are derived from KEGG database (www.genome.jp/kegg/).

### 2.11. Website Development

The current landscape of the available and emerging antiviral drug combinations was reviewed and summarized in a database that can be freely accessed at https://antiviralcombi.info. The information for the database was obtained from PubMed, clinicaltrials.gov, DrugBank, DrugCentral, the Chinese Clinical Trials Register (ChiCTR), and EU Clinical Trials Register databases [11,18,19], as well as other public sources. To extend the coverage of our database, we manually review each article to extract raw drug combination data, where available. The website was developed with PHP v7 technology using D3.js v5 (https://d3js.org/) for visualization.

## 3. Results

### 3.1. Novel Anti-SARS-CoV-2 Combinations

Currently, there is still no potently effective antiviral treatment against SARS-CoV-2. However, our previous experiments have uncovered synergism between orally available nelfinavir and several other drugs against SARS-CoV-2 in Vero-E6 cells [7]. Nelfinavir is an approved antiviral developed for use in the treatment of HIV infection, and its safety profile in humans is already understood. Nelfinavir was also shown to target the SARS-CoV-2 protease [7]. Moreover, nelfinavir has been successfully used to treat COVID-19 in several patients (https://www.researchsquare.com/article/rs-27346/v1).

We have also shown that serum samples from patients recovered from SARS-CoV-2 infections could neutralize the virus and prevent virus-mediated cell death [7]. The FDA has granted emergency use authorization for convalescent serum therapy for the treatment of COVID-19, and a recent safety study has found that its use is generally safe in patients [20].

Here, we tested nelfinavir in combination with a convalescent serum sample from a recovered patient (G614) [16]. We observed that nelfinavir and G614 serum displayed antiviral synergism without detectable toxicity in human lung epithelial Calu-3 cells (synergy score: 13; Figure 2A). Moreover, at selected concentrations nelfinavir combinations with convalescent serum G614 reduced the SARS-CoV-2 production by >2 logs in comparison to nelfinavir alone (Figure 2E)

Next, we obtained PBMCs from convalescent patients and identified their serological responses to SARS-CoV-2 antigens. We cloned antibodies using proprietary HybriFree technology [21]. We identified a panel of antibodies that bind the viral S-protein, including the 23G7 antibody, which can neutralize SARS-CoV-2 infection at nanomolar concentrations (Appendix A). When this antibody was administered to Calu-3 cells in combination with nelfinavir, we observed that the combination was synergistic and had no detectable toxicity (synergy score: 24; Figure 1).

Cycloheximide, cepharanthine, EIDD-2801, and remdesivir have been reported recently to possess anti-SARS-CoV-2 activity [22,23,24]. We confirmed their anti-SARS-CoV-2 activities on Vero-E6 cells (Appendix A). We tested combinations of cycloheximide, cepharanthine, EIDD-2801 and remdesivir with nelfinavir in Calu-3 cells. We observed that only nelfinavir plus EIDD-2801 or remdesivir were synergistic at non-cytotoxic concentrations (synergy scores: 14 and 6; Figure 2B–D). Moreover, at selected concentrations, nelfinavir with EIDD-2801 or remdesivir reduced SARS-CoV-2 production by >2 logs in comparison to nelfinavir alone (Figure 2E). Thus, the nelfinavir-convalescent serum G614, nelfinavir-23G7, nelfinavir-EIDD-2801, and nelfinavir-remdesivir combinations could result in better efficacy and decreased toxicity for the treatment of SARS-CoV-2 than drugs alone.

### 3.2. Novel Anti-EV1 Combinations

EV1 belongs to the genus *Enteroviruses*. Enteroviral infections affect humans worldwide by causing the common cold, hand-foot-and-mouth disease, meningitis, myocarditis, pancreatitis, and poliomyelitis. Enteroviruses are also associated with chronic diseases such as type I diabetes, asthma, and allergies. Enteroviruses are non-enveloped viruses that belong to the family of *Picornaviridae*. They include 12 species, enterovirus A–H, and J and rhinovirus A–C.

There are no approved therapies to treat enterovirus infections. In order to identify antiviral drug candidates, we screened the FIMM oncology drug collection (527 drugs) against EV1 in human cancer lung epithelial A549 and retinal pigment epithelial RPE cells using cell viability assay as readout. We identified vemurafenib as an inhibitor of EV1 replication. Vemurafenib (marketed as Zelboraf) is an FDA-approved inhibitor of the cellular B-Raf enzyme for the treatment of late-stage melanoma. Vemurafenib interrupts the B-Raf/MEK/ERK pathway, only if the B-Raf has V600E mutation [25]. A549 and RPE are non-BRAF mutated cells indicating that it does not target BRAF. Importantly, our studies show that vemurafenib inhibits EV1, but not EV6 (structural identity—88%, similarity—93%) in A549 cells, indicating that it could target viral protein as well (Appendix A).

Next, we studied the effect of vemurafenib on the metabolism of EV1- and mock-infected A549 cells. We analyzed 111 polar metabolites in cell culture supernatants at 24 hpi. We quantified 90 metabolites. EV1 infection affected the levels of adenine, adenosine, hypoxanthine, glutathione, NAD, AMP, guanosine, and sucrose. Vemurafenib had some effect on the levels of these metabolites in both EV1- and mock-infected A549 cells (Appendix A).

We also evaluated the effect of vemurafenib on transcription in EV-1 and mock-infected A549 cells. Cells were treated with 10 µM vemurafenib or DMSO and infected with EV1 or mock. After 8 h, we analyzed the expression of cellular genes using RNA microarray. We found that vemurafenib deregulated transcription of several genes in EV1- and mock-infected cells (Appendix A). Interestingly, gene set enrichment analysis (GSEA) revealed that vemurafenib affected transcription of 17 cellular genes belonging to GO_RESPONSE_TO_OXYGEN_CONTAINING_COMPOMPOUND and GO_RESPONSE_TO_ENDOGENOUS_STIMULUS gene sets (*p*-value 9.95 e-15 and 1.24 e-14; FDR *q*-value 9.45 e-11 and 9.45 e-11). These results indicate that vemurafenib could also target cellular factor(s) involved in the transcription of antiviral genes.

In addition, we evaluated the effect of vemurafenib on the production of cytokines and growth factors in EV1-infected and non-infected cells. After 24 h, medium from EV1- or mock-infected, DMSO- or drug-treated A549 cells was collected and clarified by centrifugation. Cytokines were analyzed using proteome profiler human cytokine array kit. Appendix A shows that EV1 replication suppressed secretion of CXCL-1, PDF-AA, CCL2, IL8, Angiogenin, IGFBP-2, and VEGF and activated production of FGF-2, whereas vemurafenib treatment reversed this virus-mediated effect.

Next, we examined whether combinations of vemurafenib with known inhibitors of EV1 infection emetine, homoharringtonine, obatoclax, gemcitabine, or dalbavancin [18] can protect cells from virus-mediated death better than vemurafenib alone. Virus- and mock-infected A549 cells were treated with an increasing concentration of vemurafenib and increasing concentrations of two drugs. After 24 h, cell viability was measured. We calculated the synergy scores as well as selectivity for each drug combinations considering their toxicity. Figure 3A,B shows that vemurafenib plus emetine, homoharringtonine, gemcitabine, or obatoclax were synergistic (synergy score > 5). However, at selected concentrations, only two combinations (vemurafenib plus emetine or homoharringtonine) reduced the EV1 production by >2 logs in comparison to vemurafenib alone (Figure 3C).

We also identified two novel agents, anisomycin and cycloheximide, which inhibited EV1 infection in A549 cells (Appendix A). We tested combinations of vemurafenib with these novel inhibitors of EV1 infection. Vemurafenib plus anisomycin or cycloheximide were synergistic (synergy score > five) and reduced the EV1 production by >2 logs in comparison to vemurafenib alone (Figure 3C). From these results, we concluded that the addition of emetine, homoharringtonine, anisomycin or cycloheximide could decrease the effective dose of vemurafenib against EV1 infection in vitro.

### 3.3. Novel Anti-HCV Combinations

Through completion of a literature review, we identified several drugs that could be combined to inhibit HCV infection in vitro [4,11,19]. We tested combinations of sofosbuvir with brequinar, emetine, homoharringtonine, and niclosamide using GFP-expressing HCV in infected Huh-7.5 cells [11]. Eight different concentrations of the compounds alone or in combinations were added to virus- or mock-infected cells. HCV-mediated GFP expression and cell viability were measured after 48 h post-infection to determine compound efficiency and toxicity. We identified two drug combinations, which lowered GFP-expression without detectable cytotoxicity at indicated concentrations: sofosbuvir-brequinar and sofosbuvir-niclosamide, with synergy scores of 24 and five, respectively (Figure 4).

### 3.4. Novel Drug Combinations against HIV-1 Infections

Through the literature review, we identified several drugs that could be combined to inhibit HIV-1 infection in vitro [4,18,19]. We tested combinations of lamivudine with brequinar, suramin, ezetimibe, minocycline, rapamycin, and monensin, as well as combinations of tenofovir with the same drugs against HIV-1-mediated firefly luciferase expression in TZM-bl cells. The firefly luciferase open reading frame is integrated into the genome of TZM-bl cells under the HIV-1 LTR promoter. Six different concentrations of the compounds alone or in combinations were added to virus- or mock-infected cells. HIV-induced luciferase expression and cell viability were measured after 48 h to determine compound efficiency and toxicity. Our screen identified two combinations (lamivudine–monensin and tenofovir–monensin) that suppressed HIV-1-mediated firefly luciferase expression without detectable cytotoxicity with synergy scores of 5.2 and 5.9, respectively (Figure 5).

### 3.5. Drug Combination Database

Thus far, research on drug synergy is often disjointed, and there has yet to be a centralized system for aggregating and summarizing synergistic, additive, and antagonistic interactions between antiviral drugs. In response to this, we have developed a freely accessible database summarizing antiviral drug combinations and their developmental status. The website is updated regularly and incorporates novel combinations as they emerge or change the statuses of existing ones as updates occur. The database comprises 985 antiviral drug combinations (Figure 6A), including two- and three-drug cocktails. It covers 612 unique drugs and 68 different viruses (Figure 6B).

To extend the coverage of our database, we manually reviewed each article for raw drug combination data that were available. We calculated synergy scores (ZIP, Bliss, Loewe, and HSA) for these. This allowed us to filter drug-pairs to select the most synergistic combinations for selected viruses and interactively investigate their raw combination data (Figure 6C). The website also includes search, filtering, graphical visualization, and download modules.

## 4. Discussion

Although the use of drug cocktails is not new, many patients and doctors still rely on monotherapies to fight viral infections [3]. Much of this is due to incomplete knowledge of drug interactions. Here, we have reported several novel combinations that show synergism and have better efficacy than single drug therapies in vitro.

In particular, we report novel anti-SARS-CoV-2 activities of nelfinavir in combination with convalescent serum or purified neutralizing 23G7 antibodies. We have also uncovered synergism between nelfinavir and orally administered investigational EIDD-2801 or intravenously administered approved-remdesivir [26,27]. Because convalescent serum and remdesivir has already been approved for the treatment of SARS-CoV-2 infection, we strongly urge further investigation into the combinations of these agents with nelfinavir. Moreover, due to the promising antiviral and neutralizing activity of EIDD-2801 or 23G7, respectively, we believe that further research on their combinations with nelfinavir is warranted.

We also identified six novel synergistic antiviral combinations with orally available anticancer vemurafenib, which successfully inhibited EV1 infection in vitro. Of six combinations, we uncovered four which stood out by inhibiting EV1 replication by over two orders of magnitude more than inhibition by vemurafenib alone. We recommend further development of vemurafenib combinations with anisomycin, homoharringtonine or emetine as anti-EV1 therapeutics.

Combination therapies are already widely used in the treatment of HCV and HIV. Our studies have identified four novel synergistic drug combinations against HCV and HIV. In the case of HCV, we combined sofosbuvir (an FDA-approved anti-HCV drug) with brequinar (an investigational anti-cancer agent) and niclosamide (an approved anthelminthic agent). In the case of HIV, we combined monensin (a veterinary antibiotic) with lamivudine and tenofovir (both approved anti-HIV agents). These combinations boosted the antiviral activity of each of the drugs.

Our study suggests that synergy is often achieved when a virus-directed drug is combined with another virus- or host-directed drug (Figure 7). This observation is in agreement with our previous studies. We propose that this schema can be used to direct the discovery of new antiviral drug combinations, although further studies are warranted to find the molecular mechanisms responsible for this synergism pattern. It is also important to note that there is a relatively poor understanding of the mechanisms that lead to synergistic interactions between the drugs. A synergy assessment that is usually performed in a mechanism-unbiased way can help to uncover the potentially synergistic combinations even without knowing their mechanisms of actions. Therefore, our drug combinations should be further studied to explain the mechanism of synergy and validated in vivo model systems.

To underscore the potential benefits and provide an organized summary of all currently known synergistic antiviral drug combinations, we constructed a web resource that can be accessed at https://antiviralcombi.info. We hope that the synergism data can be directed to further investigate the most promising and to develop the most effective antiviral combinations.

## 5. Conclusions

Here, we identified novel and reviewed known synergistic combinations against emerging and re-emerging viral infections. Our next goal is to complete preclinical studies with the most effective and tolerable combinations and translate our findings into trials in patients. These combinations may have a global impact, improving the protection of the general population from emerging and re-emerging viral infections or co-infections and allowing the swift management of drug-resistant strains. Our bigger ambition is to find algorithms for the prediction of novel antiviral drug combinations. This can be used as a means for the faster and cheaper identification of safe and effective antiviral options.

## Figures and Tables

**Figure 1 viruses-12-01178-f001:**
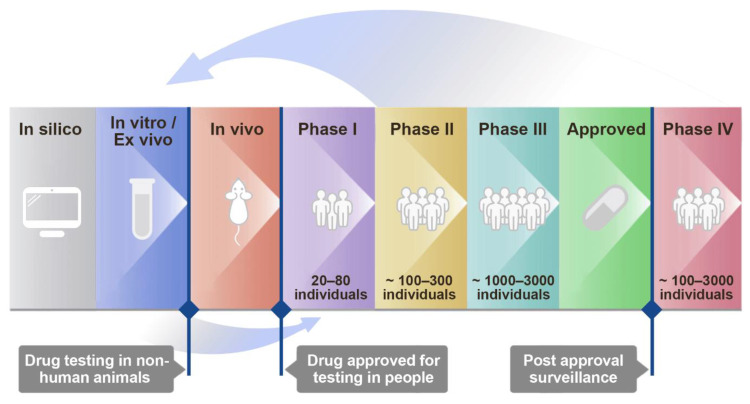
Discovery and development of antiviral drug combinations for treatment of viral infections.

**Figure 2 viruses-12-01178-f002:**
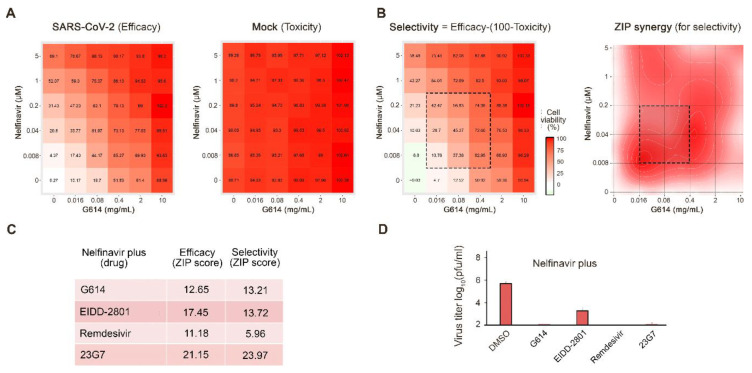
Combinations of nelfinavir with convalescent serum, neutralizing antibody, EIDD-2801, or remdesivir rescue Calu-3 cells from severe acute respiratory syndrome coronavirus 2 (SARS-CoV-2)-mediated death and inhibit virus replication. (**A**) The interaction landscapes of one of the combinations measured using a CTG assay on SARS-CoV-2- and mock-infected cells. (**B**) The interaction landscape showing selectivity and synergy of the drug combination. (**C**) Synergy scores were calculated for efficacy (SARS-CoV-2-infected) and selectivity (SARS-CoV-2-infected–Mock-infected) dose-response matrices for drug combinations. (**D**) The effects of 1 µM nelfinavir plus 0.1% DMSO, 2 mg/mL G614, 1 µM EIDD-2801, 1 µM remdesivir or 2 ng/mL 23G7 antibody on viral replication measured by plaque reduction assay.

**Figure 3 viruses-12-01178-f003:**
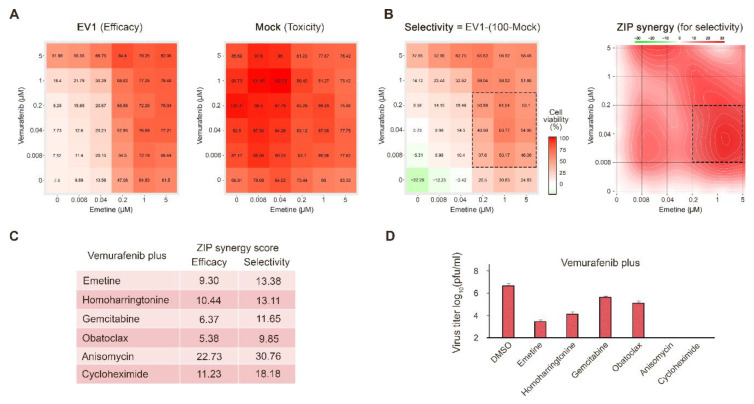
Combinations of vemurafenib with other antiviral agents rescue A549 cells from echovirus 1 (EV1)-mediated death and inhibit virus replication. (**A**) The interaction landscapes of one of the drug combinations measured using a CTG assay on EV1- and mock-infected cells. (**B**) The interaction landscape showing selectivity and synergy of the drug combination. (**C**) Synergy scores were calculated for efficacy (EV1-infected) and selectivity (EV1-infected–Mock) dose-response matrices for drug combinations. (**D**) The effects of 5 µM vemurafenib plus 0.1% DMSO, 0.2 µM emetine, 0.2 µM homoharringtonine, 0.2 µM obatoclax, 1 µL gemcitabine, 0.1 µL anisomycin and 1 µL cycloheximide on viral replication measured by plaque reduction assay.

**Figure 4 viruses-12-01178-f004:**
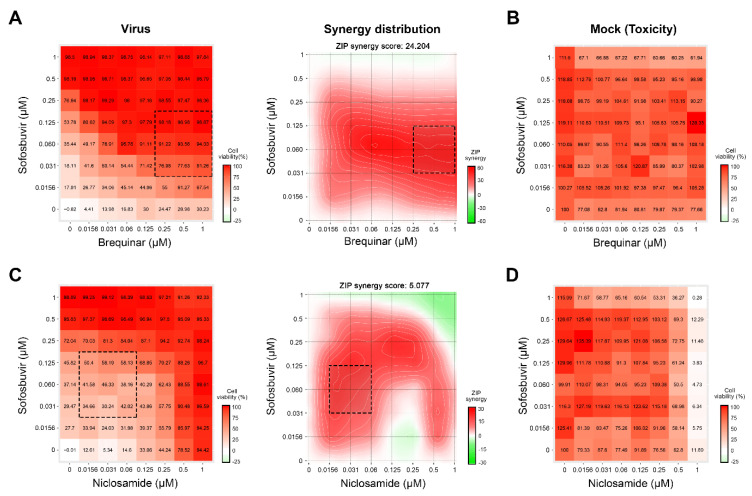
Combination of sofosbuvir and brequinar or niclosamide inhibit HCV-mediated GFP expression in Huh-7.5 cells. (**A**,**C**) The interaction landscapes measured using a GFP-expressing reporter virus. (**B**,**D**) The interaction landscapes of two drugs measured using CTG assay on mock-infected cells.

**Figure 5 viruses-12-01178-f005:**
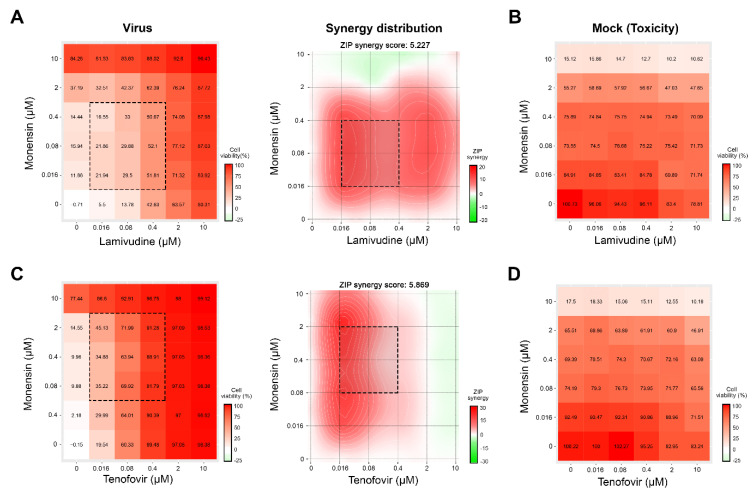
Combinations of monensin with lamivudine or tenofovir inhibit human immunodeficiency virus 1 (HIV-1)-mediated luciferase expression in TZM-bl cells. (**A**,**C**) The interaction drug combination landscapes measured as 6 × 6 dose-response matrices using an HIV-1 virus and reporter cell line expressing luciferase. (**B**,**D**) The interaction drug combination landscapes measured using and CTG assay on mock-infected cells.

**Figure 6 viruses-12-01178-f006:**
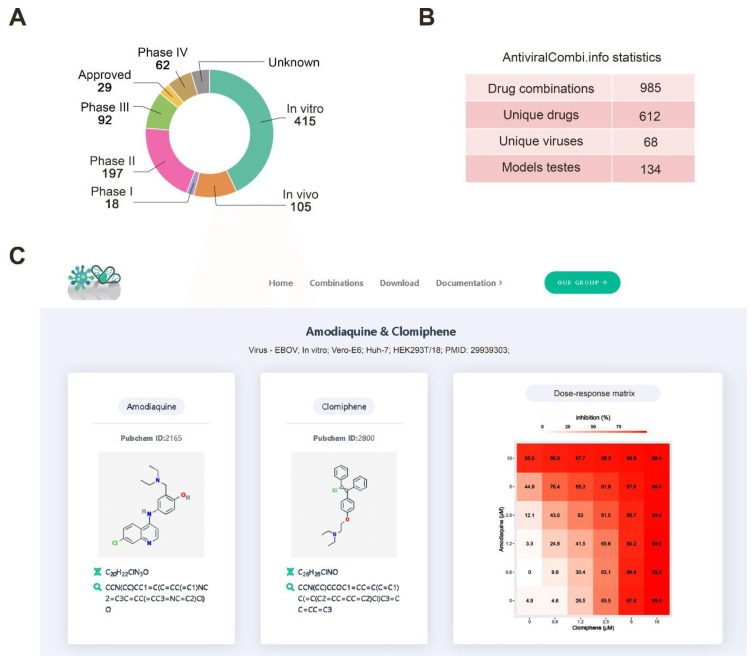
AntiviralCombi.info database summarizing existing antiviral drug combinations. (**A**) Developmental phases of combinations are presented in the database. (**B**) Database summary statistics. (**C**) An example snapshot of the database showing information on the amodiaquine and clomiphene drug combination.

**Figure 7 viruses-12-01178-f007:**
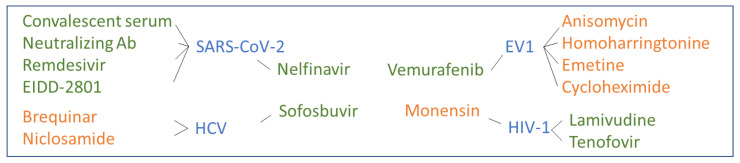
Synergistic virus- (green) and host- (orange) targeting drug combinations identified in this study. HCV: hepatitis C virus.

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
