# Peer review of "Identification and Tracking of Antiviral Drug Combinations"

_viruses, 2020, doi:10.3390/v12101178_

Round 1

Reviewer 1 Report

In the manuscript,  “Identification and tracking of antiviral drug combinations” by Ianevski et al, the investigators use combinations of possible antivirals to screen for better treatment possibilities for different viral infections.  The paper presents several different cell culture models that all arrive at the same idea that combinations of antivirals often work in synergy.  

For SAR-CoV-2 infection, use of convalescent serum along with the beta-coronavirus protease inhibitor nelfinavir along with combinations of cycloheximide, cepharanthine, and EIDD-2801 or remdesivir find combinations that show synergism in being an antiviral combination. They also test vemurafenib with other antiviral agents against enterovirus infections, which are known as not well setup with a standard of care that includes antivirals.  In addition, they test of sofosbuvir with other antiviral agents against HCV. Finally, they test monensin with lamivudine or tenofovir against HIV infection.  One point is that the display of data is useful and well organized in these figures. The antiviral database presented at the end of the article is a nice addition to their work.

The idea of repurposing these drugs is well expanded upon and currently a major focus of the race to treat viruses.  Overall, this work has some incremental findings with various combinations but presents several different models that taken together give us an interesting set of ideas and outcomes for clinical trials to setup and pursue.

Author Response

Many thanks for extremely positive feedback.

Reviewer 2 Report

This study investigated the potential advantage of combination therapies in treating emerging viruses with existing drugs. The authors tried to demonstrate  this by testing a series of combinations of different drugs in the synergistic inhibition of four pathogenic viruses, SARS-CoV-2, EV1, HCV and HIV-1. The results did show that a few effective drug combinations effectively inhibited the above viruses in cultured cells, which are worth pursuing in future studies.

The biggest challenge to formulate effective and feasible combination therapies is the potential drug interactions in vivo, including the different pharmacokinectics of the candidate drugs to be combined. Also, can any two or three drugs can be used together, regardless of their mechanisms of action? These two issues need to be adequately discussed.

Author Response

Many thanks for your positive feedback.

We now further improved the manuscript and discussed the issues associated with drug combination specificity and in vivo efficacy.

"In this study, we have identified novel and reviewed known synergistic antiviral combinations (Fig. 7). Our study suggests that synergy is often achieved when a virus-directed drug is combined with another virus- or host-directed drug. This observation is in agreement with our previous studies. We propose that this schema can be used to direct the discovery of new antiviral drug combinations, although further studies are warranted to find the molecular mechanisms responsible for this synergism pattern.

It is also important to note that there is a relatively poor understanding of the mechanisms that lead to synergistic interactions between the drugs. A synergy assessment that is usually performed in a mechanism-unbiased way can help to uncover the potentially synergistic combinations even without knowing their mechanisms of actions. Therefore, our drug combinations should be further studied to explain the mechanism of synergy and validated in vivo model systems."